# Policy Determination in Reinforcement Learning via Quantum Mechanics Analogies

## Abstract

This study explores the integration of quantum mechanics with reinforcement learning (RL) to determine policies. Leveraging the ground state eigenfunction of the Schrödinger equation, a direct analogy between RL policies and quantum particle probabilities is established. The approach demonstrates close alignment with conventional RL methods, offering potential for efficient policy determination in complex environments. The study extends beyond gridworld scenarios to solve maze navigation tasks, showcasing the versatility of the approach. Future research could explore extensions of these concepts to multi-agent RL scenarios.

## 1 Introduction

Classical reinforcement learning (RL) methods, such as those detailed by Sutton & Barto (2018), provide foundational techniques, including policy and value iteration, which have been extensively used in various applications. The application of RL in quantum physics has gained significant attention in recent years (Meyer et al., 2024). An early work of Dong et al. (2008) introduced a quantum reinforcement learning (QRL) method that combines quantum theory with RL principles. Carleo & Troyer (2017) demonstrated a RL scheme for solving the quantum many-body problem using artificial neural networks. Bukov et al. (2018) implemented RL techniques for quantum control tasks, showing comparable performance to optimal control methods. Paparelle et al. (2020) explored deep RL algorithms for digitally stimulated Raman passage enabling preparation of coherent superpositions in quantum systems. Finally, Cheng et al. (2023a) used the Deep Deterministic Policy Gradient algorithm for quantum control tasks. In the same year further improvement to QRL were made by Cheng et al. (2023b), which involved extending it to an offline setting, where the algorithm does not need to actively interact with the environment to gather new samples. Evidently, there is an emerging interest in applying RL techniques to various quantum physics problems, showcasing the potential of combining quantum principles with machine learning algorithms for advanced applications in quantum control and quantum many-body systems. In the context of applying concepts from quantum physics to reinforcement learning, Rahme & Adams (2021) utilized the partition function from statistical physics to develop a policy that considers state entropy, thereby favoring states with a greater number of potential outcomes. It seems promising to use the tools and ideas of quantum physics for application to RL.

In this work, inspiration is drawn from the intriguing behavior of a quantum particle's wavefunction, which extends across all space and with a propensity towards attracting regions. A parallel is made with RL, where an agent dynamically navigates towards states yielding higher rewards. This analogy is explored and applied to develop a novel method for solving the agent's policy.

## 2 Theoretical Background

In this section, we provide a brief introduction to two fundamental equations: the Bellman equation in RL and the Schrödinger equation in quantum mechanics. RL uses the Bellman equation to iteratively update the value function, which estimates the expected cumulative reward an agent can achieve from a given state under a policy. The Schrödinger equation describes the behavior of quantum particles in potential fields,

determining the particle's likely positions based on the potential energy distribution, with the ground state indicating the lowest energy level where the particle is most likely to be found.

## 2.1 Reinforcement Learning

RL is a computational approach to learning optimal decision-making policies in dynamic environments (Sutton & Barto, 2018). Central to RL is the concept of the value function, which represents the expected cumulative reward an agent can achieve from a given state under a certain policy. The solution to the value function under a policy can be determined by the Bellman equation (Bellman, 1957):

$$V^{\pi}(s) = \sum_a \pi(a|s) \left( \mathcal{R}_s^a + \gamma \sum_{s'} \mathcal{P}_{ss'}^a V^{\pi}(s') \right) \tag{1}$$

where $V^{\pi}(s)$ is the value of state $s$ under policy $\pi$, $\pi(a|s)$ is the probability of taking action $a$ in state $s$ under policy $\pi$, $\mathcal{R}_s^a$ is the expected immediate reward upon taking action $a$ in state $s$, $\gamma$ is the discount factor representing the importance of future rewards, and $\mathcal{P}_{ss'}^a$ is the probability of transitioning from state $s$ to state $s'$ under action $a$. The Bellman equation serves as the foundation for many RL algorithms, such as value iteration and policy iteration (Howard, 1960), which iteratively update the value function until convergence to find the optimal policy $\pi_*$.

## 2.2 Schrödinger Equation

The time-independent Schrödinger equation (Schrödinger, 1926) is $\hat{H}\Psi = E\Psi$, where the wave function $\Psi$ is an eigenfunction of the Hamiltonian operator $\hat{H}$ with corresponding eigenvalue(s) $E$. The Hamiltonian operator, for a quantum particle, can be expressed in terms of its kinetic energy $\hat{T}$ and potential energy $\hat{U}$:

$$\hat{H} = \hat{T} + \hat{U} = -\frac{\hbar^2}{2m}\nabla^2 + U(x). \tag{2}$$

The particle is attracted to more negative $U(x)$ so it is more likely to be found at position $x$ where $U$ is minimum, as shown in Fig. 1 for the case of a quantum harmonic oscillator. The probability distribution of the particle position is derived from the wave function $\Psi_n(x)$, where $n$ is an integer that corresponds to the quantized energy level $E_n$. The probability density at its lowest possible energy level $E_0$ (the ground state[1]) is then given by $P_0(x) = |\Psi_0|^2 = \Psi_0^*(x)\Psi_0(x)$.

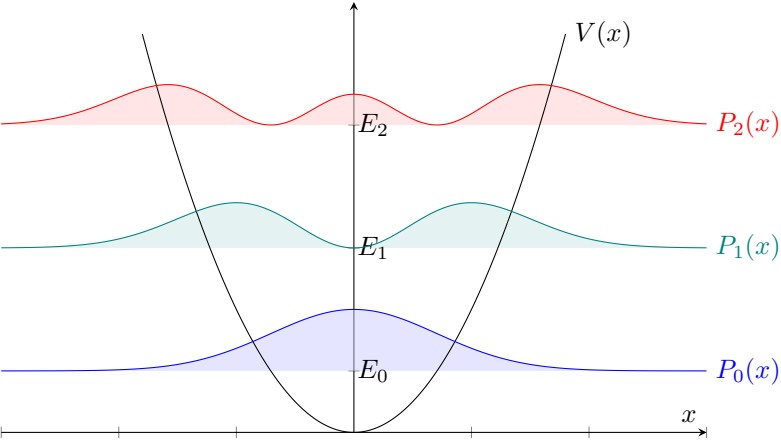

Figure 1: Probability density of the particle for the first three energy levels of the quantum harmonic oscillator. The potential is given by $U(x) = x^2$.

---

[1]Not to be confused with the term *state* used in the RL context.

# 3 Forming the Analogy Between an Agent and a Ground State Quantum Particle in a Potential

In this section, we describe how concepts from RL can be analogously applied to ground state quantum particles in a potential. RL's action-value function guides actions based on expected rewards, while the policy function selects actions maximizing rewards for each state. This analogy extends to ground state quantum particles in a potential, where a policy guides transitions between vertices in a graph. Here, the adjacency matrix and ground-state probability density determine the policy, analogous to RL's action-value function and policy.

## 3.1 Potential Function as a Reward Function

In RL, the agent's action policy is determined by the action that takes the agent from states of low value towards states of higher value — a gradient-ascent. Similarly, a particle goes from high potential towards lower potential. Therefore, the first point of analogy to RL comes from making $U \to -R$ in Eq. 2. This makes the Hamiltonian (dropping the constant) as:

$$\hat{H} = -\nabla^2 - R(x). \tag{3}$$

In this way, a particle and an agent both tend towards the trough of the function $R$. For comparison, Fig. 2 shows the solution to the optimal value function for the reward function $R = -U(x) = -x^2$.

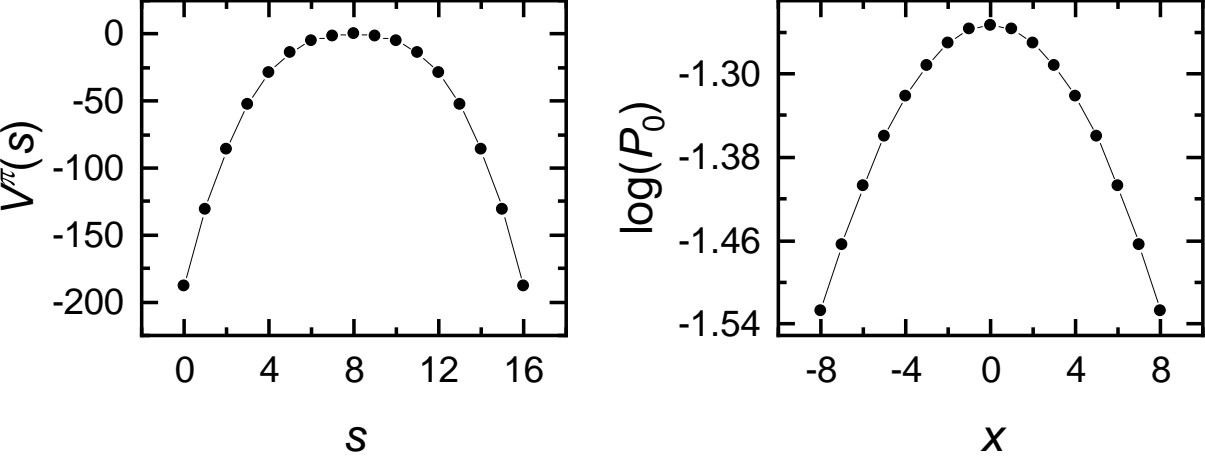

Figure 2: (left) Optimal value function for the reward function $R(s) = -(s-8)^2$ where $s \in \mathbb{Z} \cap [0, 16]$. Three possible actions make the following state-transitions $s \to s+1$, $s \to s-1$, and $s \to s$. Actions that would transition to out-of-bounds result in the same state. The closed-form solution to the undiscounted optimal value function following the optimal policy is $V(s) = -\frac{1}{6}(2|s-t|+1)(|s-t|+1)|s-t|$ where $t = 8$ is the state at $R = 0$. (right) The log-probability density of the ground state of the quantum harmonic oscillator is proportional to $-x^2$. Note that $x \in \mathbb{R}$ is unbounded but is sampled between $[-8, 8]$ for a visual comparison.

Given that there are only three possible actions, $s \to s+1$, $s \to s-1$, and $s \to s$, the optimal policy is to increment the state until the peak where the reward function is $R = 0$ and then stay in the same state, the peak, indefinitely. Evidently, the optimal value function is concave as it is increasing towards the peak from either side (Fig. 2 (left)). Similarly, the solution to the probability density of the quantum harmonic oscillator is increasing towards the center of the potential from either side (Fig. 2 (right)). Should a particle be placed away from the center there will be a force on it that moves it towards the center. The idea here is to generalize this idea and let physics determine the optimal policy by converting an RL problem into a physics problem.

### 3.2 Hamiltonian on Directed Graph

In the Hamiltonian, the Laplacian $\nabla^2$ in the kinetic energy term in three-dimensional Euclidean space is

$$\nabla^2 = \frac{\partial^2}{\partial x^2} + \frac{\partial^2}{\partial y^2} + \frac{\partial^2}{\partial z^2}, \tag{4}$$

where $\frac{\partial}{\partial x}$, $\frac{\partial}{\partial y}$, and $\frac{\partial}{\partial z}$ are the partial derivative operators with respect to $x$, $y$, and $z$ coordinates, respectively. This can not be used to solve an RL problem as states and actions are graph-like whereas spatial coordinates are continuous. However, for edge-weighted directed graphs, the discreet Laplacian can be used:

$$(\Delta_\gamma \Psi)(v) \equiv \nabla^2 \Psi(v) = \sum_{w:d(w,v)=1} \gamma_{wv}[\Psi(w) - \Psi(v)],^2 \tag{5}$$

where $\nabla^2 \Psi(v)$ is the Laplacian of the function $\Psi$ at vertex $v$, $\sum_{w:d(w,v)=1}$ denotes the summation over all vertices $w$ that are adjacent (connected by a single edge) to vertex $v$, $d(w,v) = 1$ represents the 1-edge distance between vertices $w$ and $v$, $\gamma_{wv}$ is the weight of the edge between vertices $w$ and $v$, and $\Psi(v)$ and $\Psi(w)$ are the values of the function $\Psi$ at vertices $v$ and $w$, respectively. Note that in a directed graph $d(w,v)$ does not necessarily coincide with $d(v,w)$. Substituting Eq. 5 into Eq. 3 and representing the reward function on the graph domain produces the Hamiltonian operator on a graph:

$$\hat{H} = -\Delta_\gamma + U(v) = -\Delta_\gamma - R(v), \tag{6}$$

for which the ground state eigenfunction $\Psi_0(v)$ is solved numerically when represented in matrix form.

For a quick comparison of notation lets consider the case of a 3-vertex fully connected network as shown in Fig. 3. In the left diagram the system is in the RL context where each vertex corresponds to a state $s$ and the

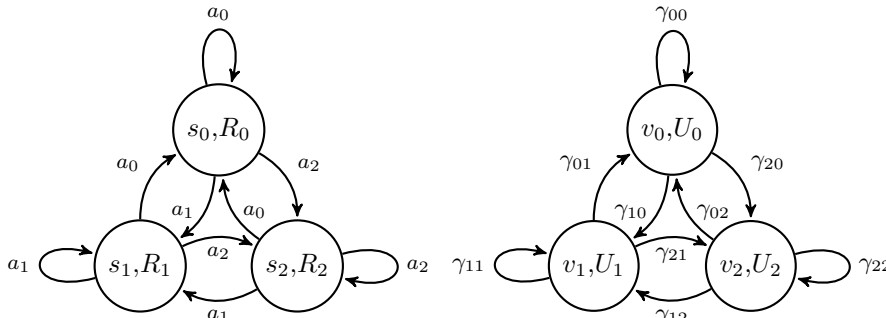

Figure 3: Comparison of a 3-vertex fully connected network in the (left) RL context and (right) physics context. There is a correspondence between states $s$ and vertices $v$, actions $a$ and edge-weights $\gamma$, and reward $R$ and potential $U$. In this example the reward obtained is independent of the action taken in a given state.

arrows correspond to transitions due to some action $a$. In this example the reward $R$ is action independent and only depend on from which state an action was done. To determine the value function would be to solve the Bellman equation (Eq. 1). For the diagram on the right the system is represented in the physics context where each vertex $v$ has a corresponding potential $U$ and the arrows determine the connectivity between the vertices. Comparing the two graphs it is easy to see the analogy: $s \Leftrightarrow v$, $a \Leftrightarrow \gamma$, $R \Leftrightarrow U$. Furthermore, the edge weight $\gamma$ in the physics context is analogous to the transition probability $\mathcal{P}_{ss'} = \sum_a \pi(a|s)\mathcal{P}_{ss'}^a$ in the RL context. The matrix form of the graph Laplacian for this example is

$$\Delta_\gamma = \begin{bmatrix} -(\gamma_{10} + \gamma_{20}) & \gamma_{10} & \gamma_{20} \\ \gamma_{01} & -(\gamma_{01} + \gamma_{21}) & \gamma_{21} \\ \gamma_{02} & \gamma_{12} & -(\gamma_{02} + \gamma_{12}) \end{bmatrix}. \tag{7}$$

---

[2] The sign is chosen to match the 2nd order finite difference, i.e. $\frac{\partial^2 \Psi(n)}{\partial n^2} = \Psi(n+1) - 2\Psi(n) + \Psi(n-1)$.

The matrix representation of the graph Hamiltonian (Eq. 6), used for calculating the ground state eigenfunction $\Psi_0(v)$, is derived by adding the elements of $U(v)$ to the diagonal of the graph Laplacian. The probability density, distributed across all vertices $v$, is $P_0(v) = |\Psi_0(v)|^2$.

### 3.3 Policy Function from Ground State Eigenfunction

In RL, the policy function $\pi(s)$ maps states to actions. $\pi(s)$ is determined by the action $a$ that maximizes the action-value function $Q^\pi(s, a)$ for that state $s$, i.e. $\pi(s) = \arg\max_a Q(s, a)$. Expanding $Q^\pi(s, a)$ in terms of $V^\pi(s)$ and $\mathcal{P}^a_{ss'}$:

$$\pi(s) = \arg\max_a(\mathcal{R}^a_s + \gamma \sum_{s'} \mathcal{P}^a_{ss'} V^\pi(s')). \tag{8}$$

Assuming that the expected immediate reward does not depend on the action taken, i.e. $\mathcal{R}^a_s = \mathcal{R}_s$, then $\arg\max_a$ only depends on the second term:

$$\pi(s) = \arg\max_a \sum_{s'} \mathcal{P}^a_{ss'} V^\pi(s'). \tag{9}$$

It can be seen that if transitions are deterministic ($\mathcal{P}^a_{ss'} = 1$ or $0$) then the policy are the actions that move the agent towards states of higher value $V^\pi(s') > V^\pi(s)$.

In the physics context, the policy is the set of transitions guiding the hypothetical particle from vertices with low probability density to those with higher probability density. Thus, in this context, the policy function determines the transitions from vertex to vertex:

$$\pi(v) = \arg\max_{v'} A_{vv'} P_0(v'), \tag{10}$$

where $A_{vv'}$ is the adjacency matrix where a non-zero element $A_{vv'}$ indicates an edge from $v$ to $v'$ and $P_0(v')$ evaluates the ground-state probability density at the vertex $v'$. Note that in this case the edge weights $\gamma_{vv'}$ should not be included in $A_{vv'}$ as $P_0(v')$ is derived using Eq. 6 which already includes $\gamma_{vv'}$.

## 4 Results

In Fig. 4, a $4 \times 4$ gridworld problem with a known solution (Sutton & Barto, 2018), is solved using value iteration with a uniform random policy. In gridworld an agent can move in the four cardinal directions and any action taken in a non-terminal state results in a negative reward. It is evident that a good policy is to take the shortest path to the nearest terminal state, which in this case is equivalent to transitioning to states with higher value $V^\pi$, in other words, the policy is greedy with respect to the value function.

The same problem is then configured in the physics-based approach, as shown in Fig. 5(left), where the gridworld is configured as a network, with edges between vertices connecting the possible states by actions. In Fig. 5(right) the log of the ground state probability density, $\log(P_0)$ is determined by solving the eigenvalue problem[3] with a potential of $U(v) = 1$ everywhere except the terminal vertices, similar to the RL problem (where $R(s) = -1$ except terminal states). The policy $\pi(v)$ is determined by Eq. 10. Comparing this result with the RL approach, note how $\log(P_0)$ and $V^\pi$ (from Fig. 4) both only increase in magnitude as it tends towards the terminal states. Consequentially, their respective policies are exactly the same, i.e. $\pi(v) = \pi(s)$. In short, the greedy policy with respect to $P_0$ of the quantum particle, which is in its ground state, is the same as that of the greedy policy with respect to $V^\pi$ as calculated from a uniform random policy.

The physics-based method can also be used to solve a maze problem, as shown in Fig. 6. The problem is setup as a network with directed edges[4], as done for the previous problem. The potential function is defined to have a negative value (or positive reward value) at the exit node and zero elsewhere. Solving for the ground state eigenfunction and determining the policy produces the path to the exit of the maze from any starting position.

---

[3]In this problem, two degenerate eigenfunctions are present: one associated with the top-left terminal vertex and the other with the bottom-right terminal vertex. The policy is determined from the sum of the probability density of these degenerate eigenfunctions.

[4]In this problem, due to symmetry of action-states, the edges can also be made undirected.

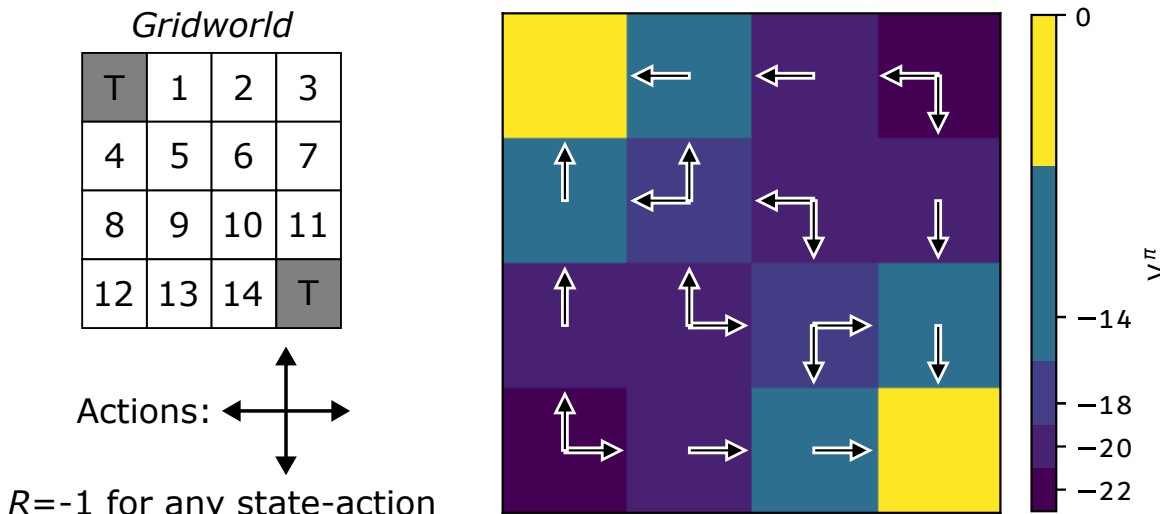

Figure 4: (left) RL environment: $4 \times 4$ gridworld. The numbers indicate accessible states and with "T" representing the terminal state. Possible actions are *left*, *right*, *up*, and *down*. Actions in nonterminal states incur a reward of -1 on all transitions. Rewards are undiscounted ($\gamma = 1$) and the task is episodic. (right) The value function $V^\pi$ is calculated by value iteration with a uniform random policy. The arrows indicate the greedy policy determined by the probability density. Note that in the terminal state a policy is not required.

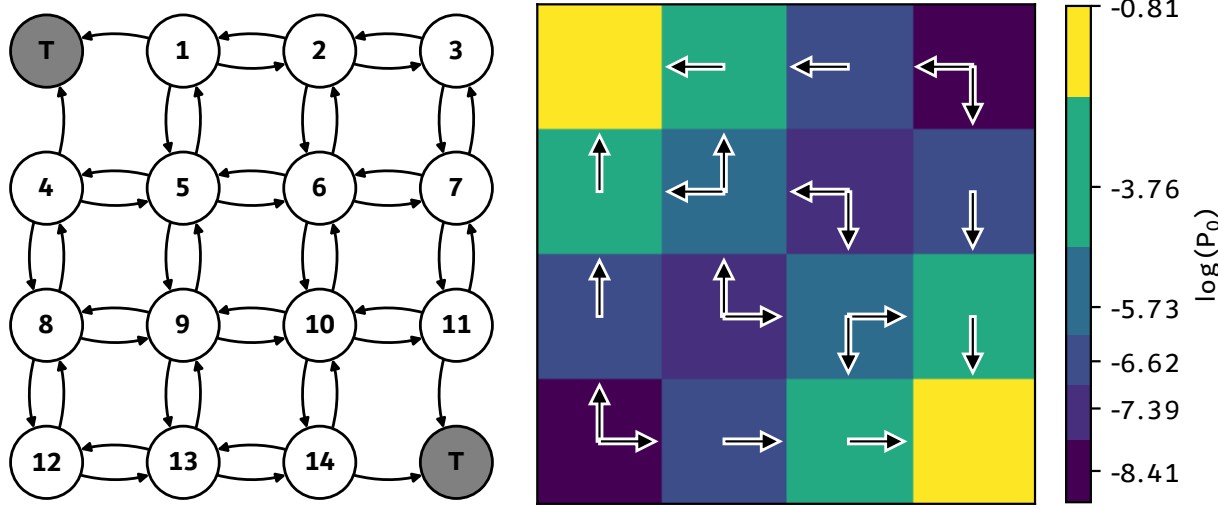

Figure 5: (left) The $4 \times 4$ gridworld problem represented as a network with nodes and directed edges. The labels inside the nodes correspond to those of Fig. 1. The potential is $U(v) = 1$ on all numbered vertices and $U(v) = 0$ on the terminal vertices. Note how the terminal vertices are evidently terminal as there are no outgoing edges. (right) The log of the ground state probability density $\log(P_0)$. The arrows indicate the action policy at the given vertex, determined by transitions towards the largest $P_0$ (of which there could be multiple). At terminal vertices there are no possible actions. Note that $P_0$ is determined from the sum of the two degenerate ground state eigenfunctions.

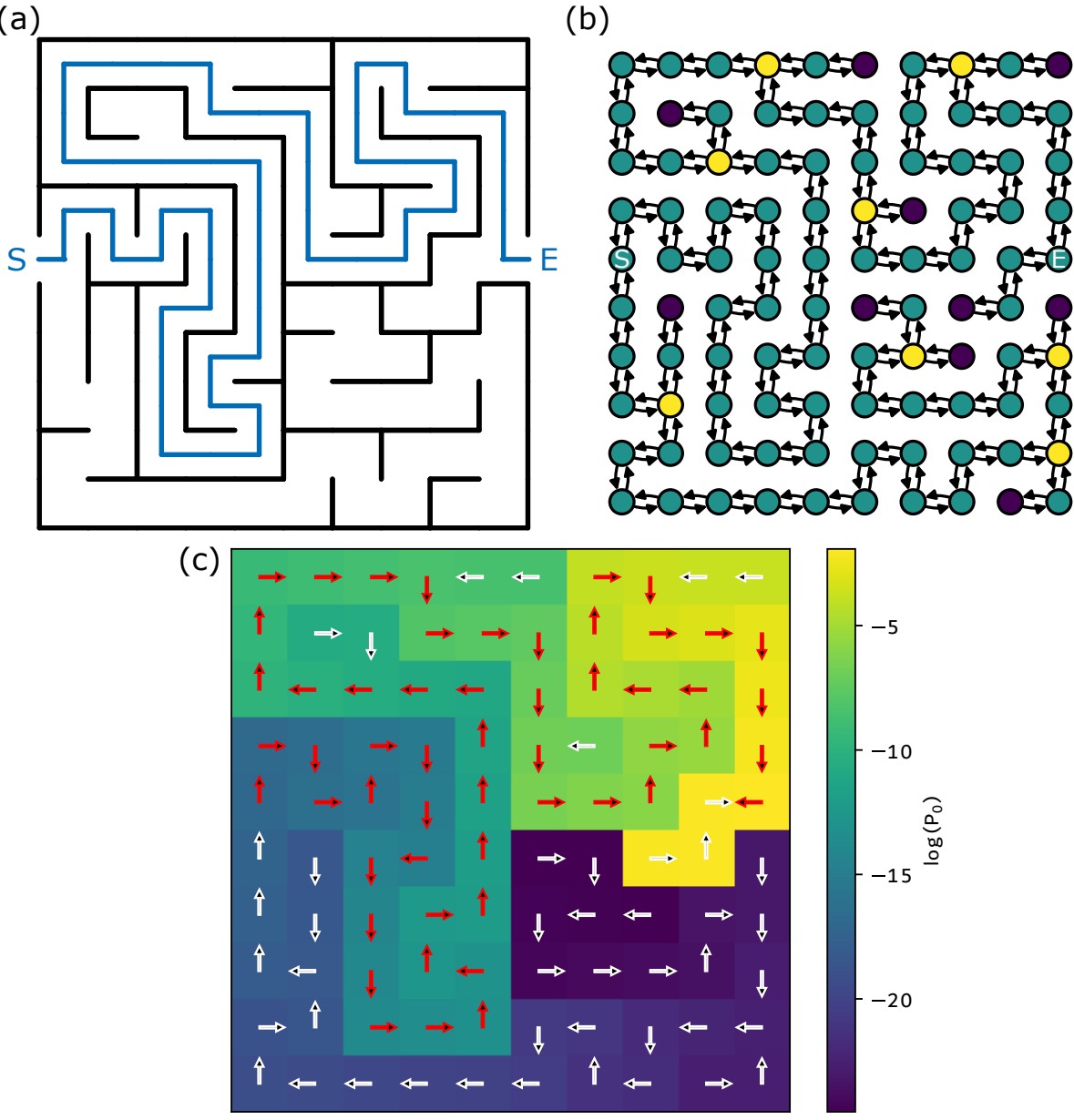

Figure 6: (a) The $10 \times 10$ maze problem with the path from the start (S) to the exit (E) of the maze indicated. (b) Network representation of the maze problem. The colors in each node represent nodes with 1 (purple), 2 (teal), and 3 (yellow) adjacent nodes, respectively. The start (S) and exit (E) node are indicated. (c) The solution to the ground state probability density. The potential is zero everywhere except $U = 1/10$ at the (E) node. The optimal policy that leads to the exit from any starting position is shown with white arrows but along the path that starts from (S) node have a red outline.

## 5 Discussion

Intuitively, it is evident that physical particles are attracted towards regions of lower energy potential, mirroring the tendency of agents in RL to seek out areas of higher reward. This is especially true of the particle with the wave function in the ground state. The excited states of the quantum particle will tend to spread its wavefunction away from the confining potential (see Appendix A.1) and thus it breaks down the

analogy with RL. Therefore, it does not appear to be useful to consider excited states for determining the policy.

Another assumption required to make a good analogy of an agent with a quantum particle is that the reward the agent receives should be independent of the action taken in a given state. The problem arises in how to encode action dependant rewards into the graph-like network used for the Hamiltonian. An approach to try might be to make additional vertices for each action but keep track of the fact they correspond to the same state.

An advantage of using the physics-based approach for determining the policy is that it does not require iterations, like in value iteration or policy iteration. This demonstrated a significant speedup (see Appendix A.2) of calculation time over the iterative algorithms, especially when the dimensionality of the problem grows with many states and/or many actions.

Using the physics-based approach, online (step-by-step) learning can be done as complete knowledge of the environment is not necessary to calculate the probability density. Consider the case that if no reward is encountered, every state has an equal probability density. Once a reward is encountered the agent is attracted towards it. To ensure all states are visited an $\epsilon$-greedy policy, i.e. greedy action with some probability, can be used. Using Fig. 5 as an example, until the agent reaches a terminal vertex and a change in potential is observed, $P_0 = 1/N$ for all nonterminal vertices visited, where $N$ is the number of states visited.

There is some flexibility in the graph approach as the Laplacian includes edge weights, and it can be reasonable to map, by analogy, the transition probability $\mathcal{P}_{ss'}$ to the edge weights $\gamma$. Consider the case when $\gamma = 0$ it would correspond to $\mathcal{P}_{ss'} = 0$ as it would imply a non-existent transition. However, $\gamma$ is bounded from 0 to $\infty$ whereas the probability upper-bound is 1. If $\gamma$ is uniform then every transition is equally likely. Take Fig. 3 as an example, going from vertex $v_0$ to $v_1$ the relative weight of that transition is $\gamma_{10}/(\gamma_{00} + \gamma_{10} + \gamma_{20})$, which would equal to $1/3$ if $\gamma_{00} = \gamma_{10} = \gamma_{20}$. Similarly, in RL, for a uniform random action policy and deterministic transitions, the transition probability going from $s_0$ to $s_1$ is $\mathcal{P}_{s_0 s_1} = 1/3$.

Excitingly, there's a potential extension of these concepts into multi-agent RL. By expanding the Hamiltonian to encompass multi-particle interactions, we open up possibilities where the state of each particle can dynamically influence the reward function of others. This could lead to intriguing developments in collaborative and competitive multi-agent systems.

## 6   Conclusion

The integration of ideas from quantum mechanics with RL offers a novel approach to policy determination. By leveraging the ground state eigenfunction of the Schrödinger equation, a direct analogy between RL policies and quantum particle probabilities is established.

The study demonstrates that the policy derived from this physics-based approach closely mirrors conventional RL methods, particularly in deterministic environments. This suggests that principles governing particle movement can effectively guide RL agents towards states of higher reward.

Moreover, the approach extends beyond gridworld or maze scenarios and can be applied to any arbitrary network. By representing RL problems as graphs and solving for ground state probabilities, policies can be derived without iterative algorithms. In future research, extending these concepts to multi-agent RL could provide interesting results and insights.

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

## A  Appendix

### A.1  Higher Energy Level States

For finding the solution to the RL problem, in the main text, only the ground state energy eigenvectors were used for calculating the probability density distribution from which the policy was derived. However, it may be interesting to consider what probability density distributions arise from higher energy level states (i.e. larger eigenvalues). Consider the small $3\times3$ gridworld problem, it has a total of 9 states and thus a total of 9 eigenvalues, as shown in Fig. 7. Note that the first two eigenvalues are exactly 0 meaning that they are degenerate. The degeneracy is due to a terminal state in the top-left corner and bottom-right corner and their probability distributions ($P_0$ and $P_1$) are mirror of the other. The policies derived from these two distributions are clearly the solutions that tend to each one of the terminal states. This is why for the full solution the probability distribution from the degenerate eigenvalues were averaged together. As the energy level $n$ increases, the probability distribution tends further and further away from the terminal states, and finally at $P_8$ it would localize the particle in the center furthest from the terminal states.

### A.2  Speed Benchmark

By modeling RL problems as graphs and determining ground state probabilities, policies can be derived without the need for iterative algorithms. To characterize the speedup in calculation time using the approach of the main text, a comparison against the value iteration methods for the $N\times N$ gridworld problem is made. The $N \times N$ gridworld problem is the same problem as in the main text Fig. 4 where there are terminal states in the top-left and bottom-right corner. However the size of the gridworld is variable by $N$, which as it increases the number of possible states grows as $N^2$. The value iteration methods with two different convergence criteria and the main text method is shown in Fig. 8.

The $\Delta$ method is where value iteration is stopped when the change in the largest value is below a threshold and the $k$ method is where iteration is stopped after a set number of iterations. The benchmark for the Schrödinger equation method considers only the time to construct the Hamiltonian and solve for the

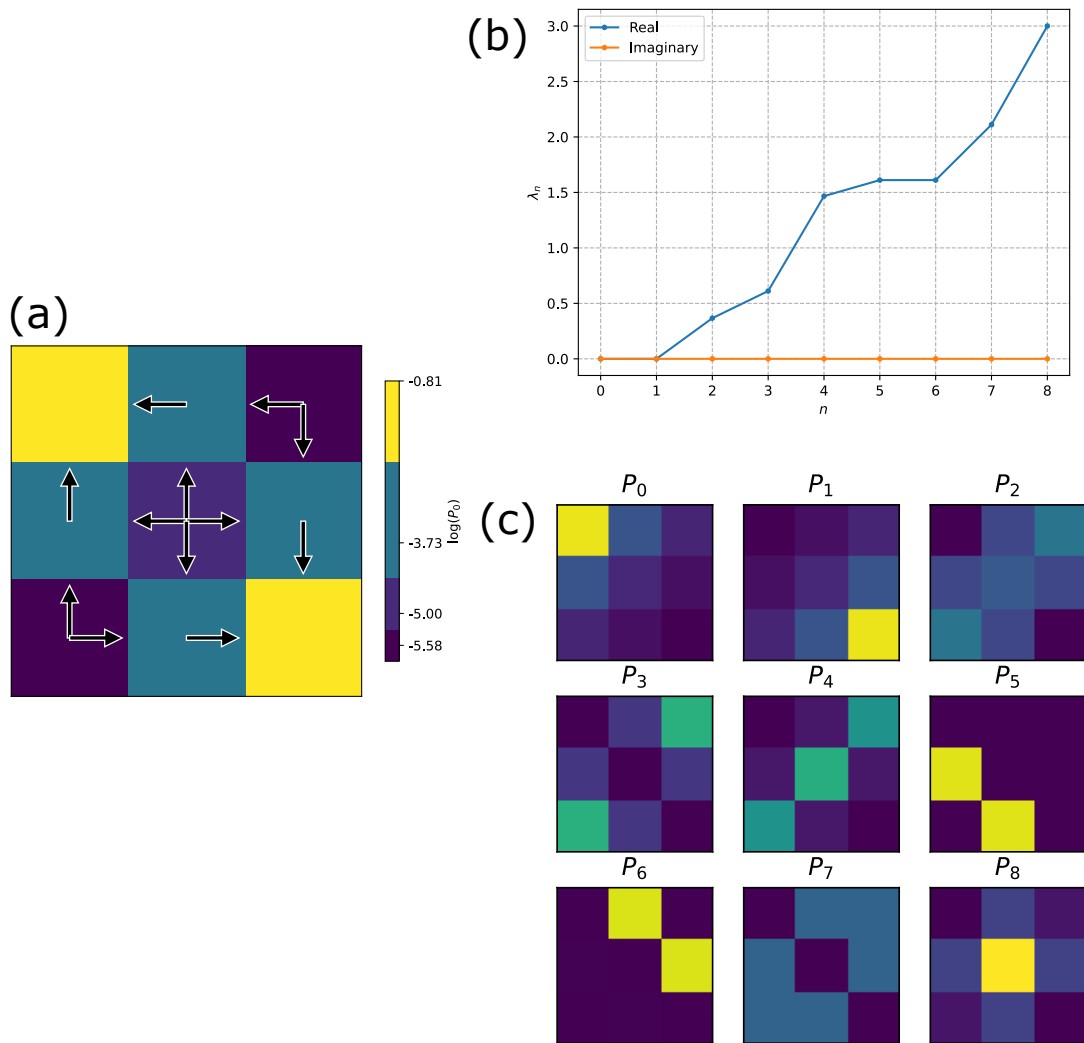

Figure 7: (a) Ground state probability of the $3\times3$ gridworld problem solved by the Schrödinger equation method and the policy indicated by arrows. (b) The real and imaginary component of the eigenvalues ($\lambda_n$) determined by the Hamiltonian describing the $3\times3$ gridworld problem. The $n = 1$ and $n = 2$ eigenvalues are equal to 0. The imaginary components are all 0. (c) All 9 of the probability distributions corresponding to the 9 eigenvalues. The $P_0$ and $P_1$ distributions correspond to the $\lambda_0 = \lambda_1 = 1$ degenerate eigenvalues.

eigenvalue and eigenvectors. As seen in Fig. 8, for most $N$ used in the benchmark there is a clear 4 order of magnitudes faster calculation time using the Schrödinger equation method as compared to value iteration. The growth rates of calculation time for the $\Delta$ and $k$ methods are about order of 6.5 and 4, respectively. Generally, the $\Delta$ method requires more iterations to reach the same threshold value as the size of the problem grows. Interestingly, the Schrödinger equation method grows at a rate of about order of 5, which is larger than the $k$ method but more importantly - smaller than the $\Delta$ method. The $k$ method can be disregarded past about $N$=6 as the solution does not produce a convergent policy whereas the $\Delta$ method gives the expected result.

The code for benchmarking ran on Python 3.12 using scipy v1.14.1 for the eigenvalue calculation used in the Schrödinger equation method. The function for value iteration ($\Delta$ method) was done as follows:

```python
import numpy as np
S = range(N_states)
A = range(N_actions)
V = np.zeros(N_states)
R = np.unique([r_func(s,a) for s in S for a in A])
Delta = np.inf
while Delta>Delta_cutoff:
    Delta = 0
    for s in S:
        v = V[s]
        V[s] = sum([Pi(s,a)*sum([p_func(sp,r,s,a)*(r+gamma*V[sp]) \
                                for sp in S for r in R]) for a in A])
        Delta = max(Delta, abs(v - V[s]))
```

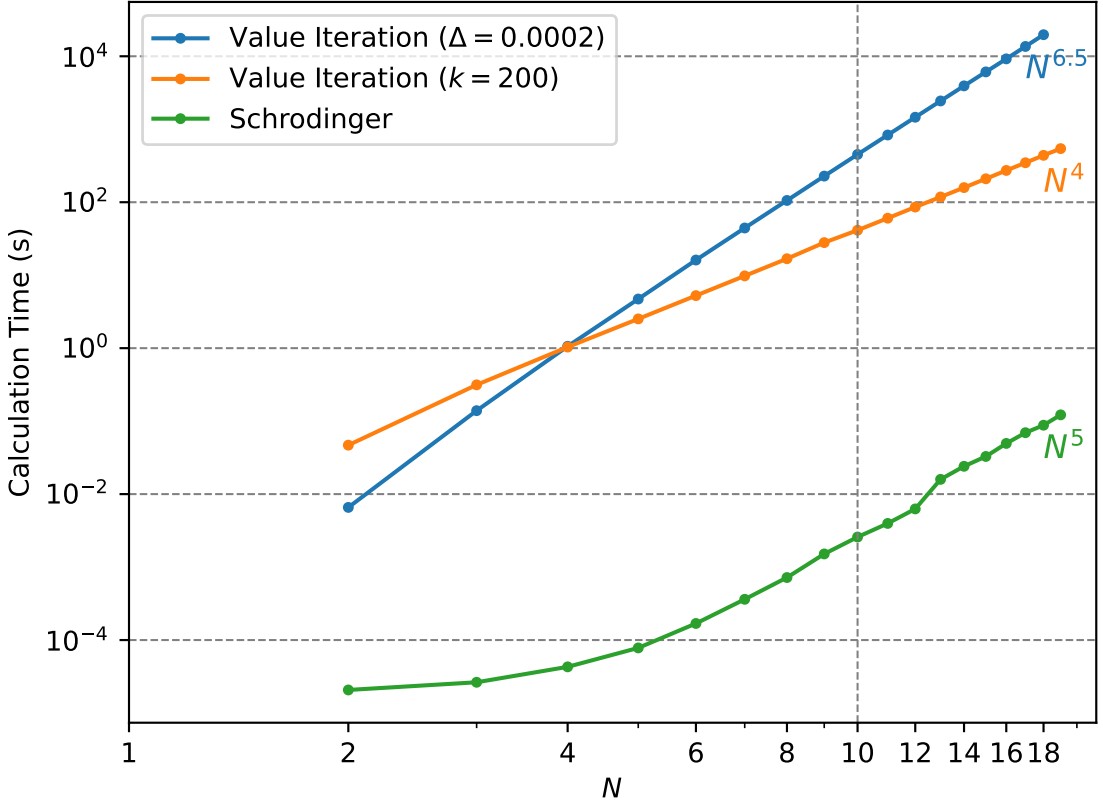

Figure 8: Time to calculate the value function versus the size of a $N \times N$ gridworld problem. The blue and orange lines are for value iteration using the $\Delta$ and $k$ convergence criteria, respectively. The green line is solving for the probability density by using the Schrödinger equation and solving for the eigenvector (i.e. the method described in the main text). The $N^{\#}$ label next to the lines indicate the growth rate of the calculation time with the size of $N$. Note that the total number of states (or nodes) is equal to $N^2$.

The function `r_func(s,a)` and `p_func(sp,r,s,a)` is the reward structure and transition probabilities of a $N \times N$ gridworld problem, respectively. The policy function `Pi(s,a)` is merely equal to 1 normalized by the number of actions $(1/N_{\text{actions}})$. The $k$ method merely replaces the while loop with a for loop with $k$ iterations.

