# OpenReview forum: "Policy Determination in Reinforcement Learning via Quantum Mechanics Analogies"
_TMLR — Rejected by TMLR_

### Review · Reviewer_VeK8 · 2024-09-05

**Summary Of Contributions:**

The paper introduces an analogy between the solution of the time-independent Schrödinger equation, which describes a quantum particle in an arbitrary potential $V(x)$ with $x$ a position co-ordinate, and the greedy policy update rule of reinforcement learning according to Bellman's expression. The basic analogy is that when transition probabilities are known exactly, one can encode them in the Laplacian operator (which describes the kinetic component of the particle's Hamiltonian), which can then be applied to a state which analogises the value function. Since the ground state of this new operator is the optimal value function, the greedy action is the minimiser of the resulting optimisation problem, yielding the greedy policy over the state space.

**Audience:**

No

**Broader Impact Concerns:**

No broader impact concerns.

**Claims And Evidence:**

Yes

**Requested Changes:**

I request the authors to find at least one angle where the connection described gives something new atop existing literature. This will be essential for my recommendation of acceptance.

**Strengths And Weaknesses:**

Strengths:
- The analogy is well described and elaborated upon.
- The mathematics appears to be correct.
Weaknesses:
- The method described is not at all novel, and yields only a basic version of very well established Dynamic Programming methods for RL.
- There are no actual experiments with quantum particles in a discrete environment; I think this would at least give some experimental curiosity.
- It is stated at the end that online learning can be used in the physics-based context, which is not surprising, since step-wise learning is already well understood (in fact, this makes up an entire subfield of RL). The physics-based approach appears not to add anything to this.

In summary, I do not think this work adds anything to the existing literature in machine learning.

---

> ### Author Response · Authors · 2024-10-07
> **Response**
>
> Dear reviewer, thank you for your time and feedback. I also appreciate the succinct summary and correct understanding of the paper. In the following I will address your comments.
> > * The method described is not at all novel, and yields only a basic version of very well established Dynamic Programming methods for RL.
>
> The goal was to first show the correspondence of the quantum mechanics derived policies with the well established DP methods for RL. I believe the method is novel as I could not find another work mapping the reward function to the potential of a Hamiltonian.
>
> > * There are no actual experiments with quantum particles in a discrete environment; I think this would at least give some experimental curiosity.
>
> Discrete-time quantum-walk on an atomic lattice via a distance-selective spin-exchange interaction (M. Khazali, Quantum 6, 664 (2022)) seems to be a promising step towards discrete quantum systems. Moreover, adapting to the case of continuous space is as simple as using the regular Schrodinger equation without adapting it to graphs.
>
> > * It is stated at the end that online learning can be used in the physics-based context, which is not surprising, since step-wise learning is already well understood (in fact, this makes up an entire subfield of RL). The physics-based approach appears not to add anything to this.
>
> For online learning, due to the large speedup in calculation (please see appendix in revised pdf), every state can be rapidly updated per step even for very large state/action-space.

---

> > ### Comment · Reviewer_VeK8 · 2024-10-25
> > **Response**
> >
> > I appreciate the author's effort in addressing my concerns.
> >
> > I still do not quite agree that a mathematical similarity/equivalence constitutes a novelty. Hamiltonians are operators on wave-vectors, which need only belong to a Hilbert space, so it is not entirely surprising that a cost function over a grid environment can be written within this expressive space (Hamiltonians are, after all, a type of cost functional).
> >
> > Having said that, the speedup is quite interesting, although a little difficult to explain from what is written (and non-asymptotic, so it's even harder to discern whether the discrepancy in the constant is not a detail of the implementation without a more complete characterisation).
> >
> > With regard to online learning, except for the speed-up within a certain regime, I don't quite see the connection. Online learning with a complete state-action space is not too hard, and I don't see any more discussion on this front.

---

### Review · Reviewer_b4uE · 2024-09-16

**Summary Of Contributions:**

The submission aims to show a theoretical connection between quantum mechanics (Schrödinger equation) and dynamic programming (Bellman equation) and to calculate reinforcement learning policies using techniques known from quantum mechanics.

Remark: \
Since no reference is made to quantum computing, this is not quantum reinforcement learning.

**Audience:**

Yes

**Claims And Evidence:**

Yes

**Requested Changes:**

see "Weaknesses", "In detail:", and "Further remarks:"

**Strengths And Weaknesses:**

**Strengths**
* The approach is original.
* The text is comfortable to read.

**Weaknesses**
* The idea is only touched on and not dealt with comprehensively. The reader is left with many questions. This may be appropriate for a workshop paper, but in my opinion it is unsatisfactory for a journal paper.
* The title seems misleading to me. With “quantum inspired” I would have expected concepts that have emerged from quantum computing and not those that are known from quantum mechanics.
* The topic of quantum reinforcement learning (QRL) is briefly touched on in the introduction, but the distinction from QRL is not made sufficiently clear.
* In my opinion, the functionality is not presented in sufficient detail.
* The potential benefits remain unclear.


**In detail:**
* The objective of using methods known from physics to demonstrate an alternative approach to reinforcement learning was undertaken for statistical physics in [1]. I am convinced that the presentation chosen there can provide the authors with helpful suggestions.
* The motivation given in the introduction needs to be clarified. As correctly stated, (Dong et al. 2008) is an early work on QRL. Accordingly, it also deals with quantum computing and qubits. The field of QRL has already gained considerable attention, see e.g. the 2022 survey [2] and the references therein. After 2022, QRL has been extended to the offline setting [3],[4],[5].
After (Dong et al. 2008) was briefly mentioned, examples are given that use RL to optimize quantum systems. Overall, the motivation remains unclear. In particular, sentence “However, … the application of quantum physics to reinforcement learning is not as thoroughly investigated.“ is not convincing, as there is extensive literature on QRL, of which only one paper is mentioned. If the distiction between QRL (using qubits) and "quantum physics" is intended, then this should be explained and emphasized in the text.
* The term quantum-inspired is used only in the title, without explaining what is meant by it or providing a reference (unfortunately, I have no specific recommendation for the references). Alternatively, the term can probably be dispensed with; in the narrower sense, it probably does not fit so well because no concepts of quantum computing are used here.

**Further remarks:**
* The abbreviation RL is introduced 5 times. Does it really have to be that often?
Sometimes “reinforcement learning” is capitalized, sometimes it is written in lower case. This should be handled consistently.
* The abbreviation RL is used in the introduction without having been introduced. The fact that it was introduced in the abstract does not count, as the main text must also be readable without the abstract.
* “a RL” -> ”an RL”
* “for which the ground state eigenfunction Ψ0(v) is solved for numerically” -> ”for which the ground state eigenfunction Ψ0(v) is solved numerically”
* In the references there are unintentional lower case letters, “raman”, “eigenwertproblem”

[1] J. Rahme and R.P. Adams, A Theoretical Connection Between Statistical Physics and Reinforcement Learning, 2021\
[2] N. Meyer et al., A survey on quantum reinforcement learning, 2022\
[3] Z. Cheng et al., Offline Quantum Reinforcement Learning in a Conservative Manner, 2023\
[4] M. Periyasamy et al., BCQQ: Batch-Constraint Quantum Q-Learning with Cyclic Data Re-uploading, 2024\
[5] S. Eisenmann et al., Model-based Offline Quantum Reinforcement Learning, 2024

---

> ### Author Response · Authors · 2024-10-07
> **Response**
>
> Thank you for your insightful feedback and the time you dedicated to reviewing my manuscript. Your comments are very helpful in improving the quality of the work. In the following I will address your comments.
> > * The title seems misleading to me. With “quantum inspired” I would have expected concepts that have emerged from quantum computing and not those that are known from quantum mechanics.
>
> Thank you for pointing out the potential misunderstanding. I have changed the title to “Policy Determination in Reinforcement Learning via Quantum Mechanics Analogies” in order to emphasise the analogy is with quantum mechanics rather than specifically quantum computing.
>
> > * The potential benefits remain unclear.
>
> Although previously I have merely described the speedup in the discussion, in this revision I have added an appendix showing the 4 order of magnitude speedup of using the quantum mechanics based approach versus the traditional value iteration. Also in terms of potential benefits is that it could be extended to a multi-particle and thus by analogy to a multi-agent scenario.
>
> > * The motivation given in the introduction needs to be clarified. As correctly stated, (Dong et al. 2008) is an early work on QRL. Accordingly, it also deals with quantum computing and qubits. The field of QRL has already gained considerable attention, see e.g. the 2022 survey [2] and the references therein. After 2022, QRL has been extended to the offline setting [3],[4],[5]. After (Dong et al. 2008) was briefly mentioned, examples are given that use RL to optimize quantum systems. Overall, the motivation remains unclear. In particular, sentence “However, … the application of quantum physics to reinforcement learning is not as thoroughly investigated.“ is not convincing, as there is extensive literature on QRL, of which only one paper is mentioned. If the distiction between QRL (using qubits) and "quantum physics" is intended, then this should be explained and emphasized in the text.
>
> The sentence was misleading as there is indeed extensive literature on QRL so I have removed it. I also have added some of the references you mentioned to the introduction section.
>
>
> > * Sometimes “reinforcement learning” is capitalized, sometimes it is written in lower case. This should be handled consistently.
>     “a RL” -> ”an RL”
>     “for which the ground state eigenfunction Ψ0(v) is solved for numerically” -> ”for which the ground state eigenfunction Ψ0(v) is solved numerically”
>     In the references there are unintentional lower case letters, “raman”, “eigenwertproblem”
>
>  Thank you, I have made the corrections.

---

> > ### Comment · Reviewer_b4uE · 2024-10-17
> > **Still some work to do**
> >
> > In my opinion, changing the title is an improvement that creates more clarity.
> >
> > The introduction is not quite round yet and should also be revised for clarity.
> >
> > I think the statement “The application of RL in quantum physics has gained significant attention in recent years” cannot be substantiated by (Meyer et al., 2024), which is a survey on quantum RL. Perhaps quantum RL should not be mentioned at all because the present work not about quantum RL. If quantum RL is still to be mentioned, then it must be made clearer that this is not about quantum RL.
> >
> > The sentence “In the context of applying concepts from quantum physics to reinforcement learning, Rahme & Adams (2021) utilized the partition function from statistical physics to develop a policy that considers state entropy, thereby favoring states with a greater number of potential outcomes.” is also not convincing. It could simply be omitted.
> >
> > I think that to get more clarity, a Related work section should be included, where  Rahme & Adams (2021) could then be mentioned, because it is related in the sense that also insights for RL are sought with concepts from physics.
> > Perhaps the Related work section would also be the place to discuss quantum RL.
> >
> > “a RL” -> “an RL”

---

### Review · Reviewer_b9CR · 2024-09-25

**Summary Of Contributions:**

This manuscript would like to draw some connection between Schrödinger equation with reinforcement learning.

**Audience:**

No

**Broader Impact Concerns:**

N/A.

**Claims And Evidence:**

No

**Requested Changes:**

I don’t think this manuscript is suitable for publication, even after major revision.

**Strengths And Weaknesses:**

### Weaknesses:
* Most of this manuscript is not even wrong: there are no discussion on why we should try to connect these two fields: no unified views, no takeaways, no actionable items. I don’t know what can and should we get from this manuscript,
* The only connection the authors want to make is between reward function and potential function. However, lots of the question remains: what’s the meaning of other reinforcement learning stuffs and quantum mechanics stuffs? For example, what is the corresponding concepts in RL on excited states/energy level/excitation etc.? I don’t see any discussion on this. I would also personally think value function is more like the potential function and reward functions are the energy that we require to excite to the high energy-level excited states. But anyway there are no discussions in this manuscript.
* I don’t know what’s the meaning of the results, given there are no accurate correspondence and no new methods.

---

> ### Author Response · Authors · 2024-10-07
> **Response**
>
> Thank you for your thoughtful feedback. I acknowledge the need to better articulate the motivation behind connecting quantum mechanics and reinforcement learning concepts. In the following I will address your comments.
> > there are no discussion on why we should try to connect these two fields: no unified views, no takeaways, no actionable items. I don’t know what can and should we get from this manuscript,
>
> The core idea of my work is to draw a novel analogy between the potential energy in quantum mechanics and the reward function in RL, aiming to explore new ways of optimising policy through this perspective.
>
> > what’s the meaning of other reinforcement learning stuffs and quantum mechanics stuffs? For example, what is the corresponding concepts in RL on excited states/energy level/excitation etc.? I don’t see any discussion on this.
>
> The higher energy states create eigenvectors and therefore probability density distributions that tend further away from the attracting potential. In the previous revision I had touched on this in the discussion, however, please see the appendix in the revised manuscript for a figure showing this effect in more detail for an easy to understand example. Policies derived from these excited/higher level energies will cause the agent to increasingly avoid the terminal states. Furthermore, it may be of interest to you that there is a large 4 orders of magnitude speedup using this method compared to traditional value iteration (see appendix in revised manuscript).

---

### Decision · Action_Editor_LnsR · 2024-10-25

**Recommendation:** Reject

**Comment:**

While the reviewers found some aspects of the paper somewhat interesting, all reviewers found the paper far too preliminary for this venue. The reviewers could not identify consistent evidence for a clear contribution that would support the claims made in the paper. It seems like the paper would need substantially more experiments to demonstrate the utility of the connection — e.g. experimenting with more complex environments and ensuring that baseline comparisons are fair (for example, that the speedup claims don't rely on giving extra information about the graph structure to the proposed algorithm relative to the value iteration baseline). Thus, I would encourage the authors to develop this project more deeply. I think that a major revision that enhances the work with deeper experiments, e.g. testing against some standard RL benchmarks (gym or atari) to show that the method is not restricted to toy settings and showing speedups relative to state-of-the-art algorithms in them, could be resubmitted in the future. Otherwise, I'd encourage the authors to submit work at this stage to a relevant workshop.

**Audience:**

Maybe in a more developed version (see comment).

**Claims And Evidence:**

No (see comment)

**Resubmission Of Major Revision:**

The authors may consider submitting a major revision at a later time.